# Adult mouse dorsal root ganglia neurons form aberrant glutamatergic connections in dissociated cultures

**F. Kemal Bayat[1,2], Betul Polat Budak[3,4], Esra Nur Yiğit[3,5], Gürkan Öztürk[3], Halil Özcan Gülçür[1,4]\*, Albert Güveniş[1]**

**1** Institute of Biomedical Engineering, Bogazici University, İstanbul, Turkey, **2** Department of Electrical and Electronics Engineering, Faculty of Engineering, Marmara University, İstanbul, Turkey, **3** Regenerative and Restorative Medicine Research Center (REMER), Research Institute for Health Sciences and Technologies (SABITA), Istanbul Medipol University, İstanbul, Turkey, **4** Faculty of Engineering and Natural Sciences, Biruni University, İstanbul, Turkey, **5** Institute of Biotechnology, Gebze Technical University, İzmit, Turkey

\* gulcur@boun.edu.tr

**Data Availability Statement:** All relevant data are within the paper and its Supporting Information files.

## Abstract

Cultured sensory neurons can exhibit complex activity patterns following stimulation in terms of increased excitability and interconnected responses of multiple neurons. Although these complex activity patterns suggest a network-like configuration, research so far had little interest in synaptic network formation ability of the sensory neurons. To identify interaction profiles of Dorsal Root Ganglia (DRG) neurons and explore their putative connectivity, we developed an in vitro experimental approach. A double transgenic mouse model, expressing genetically encoded calcium indicator (GECI) in their glutamatergic neurons, was produced. Dissociated DRG cultures from adult mice were prepared with a serum-free protocol and no additional growth factors or cytokines were utilized for neuronal sensitization. DRG neurons were grown on microelectrode arrays (MEA) to induce stimulus-evoked activity with a modality-free stimulation strategy. With an almost single-cell level electrical stimulation, spontaneous and evoked activity of GCaMP6s expressing neurons were detected under confocal microscope. Typical responses were analyzed, and correlated calcium events were detected across individual DRG neurons. Next, correlated responses were successfully blocked by glutamatergic receptor antagonists, which indicated functional synaptic coupling. Immunostaining confirmed the presence of synapses mainly in the axonal terminals, axon-soma junctions and axon-axon intersection sites. Concisely, the results presented here illustrate a new type of neuron-to-neuron interaction in cultured DRG neurons conducted through synapses. The developed assay can be a valuable tool to analyze individual and collective responses of the cultured sensory neurons.

## Introduction

Sensory neurons innervate internal and external organs and transmit noxious and non-noxious information to the Central Nervous System (CNS). They are pseudo-unipolar cells with

**Funding:** This work is funded by Boğaziçi University Research Fund to author AG under the Project Code 8080D. During the experiments some facilities of REMER (Istanbul Medipol University, Regenerative and Restorative Medicine Center) were used. The funders had no role in study design, data collection and analysis, decision to publish, or preparation of the manuscript.

**Competing interests:** The authors have declared that no competing interests exist.

axons bifurcating into two distinct branches, one extending to peripheral receptors and the other to the spinal cord [1]. Action potentials generated in sensory receptors travel from the peripheral to central processes of Dorsal Root Ganglia (DRG) neurons, without passing through any synaptic connections [2–4]. Sensory neuron bodies located in the DRG are thought to behave like rarely depolarizing passive units [5], where afferent signals bypass the neuron bodies and continue to the CNS [6]. In contrast, DRG neurons typically exhibit complex neuron-to-neuron interactions and ectopic discharges emerge from their somata in cases of injury, inflammation, or strong excitation [3]. Despite being studied extensively in many platforms within varying contexts, the reason for these neuron-to-neuron interferences inside a ganglion is not clearly understood. To our knowledge, very little research so far had been carried out concerning synaptic formation and network development potentials of sensory neurons. In this work, we wanted to examine whether DRG neurons can develop functional connections through synapses and form circuits with each other in vitro. For this purpose, we studied response profiles and communication among DRG neurons in vitro using a multi-modal approach. We used local extracellular electrical stimulation through MEA electrodes and GECI-based calcium monitoring, simultaneously.

Since most DRG neurons are excitatory and glutamate is a major excitatory transmitter in peripheral and central nervous systems [7], we examined neuronal communication through glutamatergic synaptic function. Consequently, we used adult DRG neurons of a custom double transgenic mouse model, expressing GCaMP6s at its glutamatergic neuron bodies. We employed modality-free electrical stimulation and visualized spontaneous and stimulus-evoked calcium ($Ca^{2+}$) activity at cell bodies of GCaMP6s expressing DRG neurons via confocal microscopy. An almost single-cell level stimulation was achieved by adjusting the confluency of cultures grown on MEAs.

In vitro models, allowing visualization and manipulation of both neuronal and glial cells, are potent tools for providing information at various scales. The use of MEAs allows network level investigations of neuronal populations [8–10]. However, in DRG culture assays incorporating MEA platforms, certain limitations arise. The DRG neurons exhibit spontaneous activity in sub-threshold voltage levels which is hard to decipher with standard extracellular measurements. Cytokines and growth factors have been used conventionally to sensitize neural populations [11, 12] to obtain spontaneously active DRG neurons, however our multi-modal approach does not require additional factors or cytokines. In addition, calcium imaging enables observation of sub- and supra-threshold calcium transients with high spatial resolution. Genetically encoded calcium indicators (GECIs) enable monitoring calcium dynamics of neuronal populations for a theoretically unlimited amount of time, in a non-invasive fashion with high SNR [13, 14]. The GECI constructs can be targeted to specific cell types [15].

In this study we present an experimental approach for identifying interaction profiles of DRG neurons in an *in vitro* setting. We describe, for the first time, a synaptic network formation in cultured DRG neurons and demonstrate that synaptic formation has an important role in the emergent correlated activity. Our results indicate that the correlated activity is totally suppressed by post-synaptic glutamatergic antagonists. When immunocytochemistry (ICC) is applied, synapse formation is further demonstrated by the presence of presynaptic protein marker, synaptophysin. Synapses are observed mainly in axonal terminals, axon-soma junctions, and axon-axon intersection sites. Understanding the neuron-to-neuron interaction mechanisms as described here will improve our perception on sensory neuron functioning and may lead to new, effective clinical and pharmacological studies on sensory neuron disorders.

## Methods

### Ethics statement and animal handling

Transgenic mice strains were kept and bred in *The Experimental Animal Center of Istanbul Medipol University*. All animals were handled in strict accordance with guidelines for animal care and use issued by the EU directive code; 86/609/CEE. *The Committee on Ethics of Animal Experimentation of Istanbul Medipol University (IMUHADYEK)* approved all procedures.

Two transgenic mice strains purchased from Jackson Laboratories were used. The first one was a knock-in strain Vglut2-ires-cre (C57BL/6J), having Cre-recombinase enzyme expression in excitatory glutamatergic neuronal cell bodies. The second was Ai96 (RCL-GCaMP6s), a Cre-dependent calcium indicator strain (C57BL/6J), which emits EGFP fluorescence after calcium binding. These two original strains were crossbred and the offspring successfully expressed GCaMP6s in their glutamatergic neurons. In total, six transgenic animals were used in the present work.

### Dissociated DRG culture protocol

The dissociated adult DRG culture protocol was adapted from a previously published work [16]. Prior to the dissection, mice were euthanized with $CO_2$ asphyxiation and rapid decapitation. In a sterile hood, ganglia were collected in the *Dissection Medium*. Collected ganglia were transferred into the *Enzyme Solution 1*, and incubated in 37˚C, 5% $CO_2$ for 40 minutes. Then, the ganglia were washed with Hank's Balanced Salt Solution, (HBSS, Sigma) and transferred into *Enzyme Solution 2* for 15 minutes at 37˚C, 5% $CO_2$ incubation. Following the incubation in *Enzyme Solution 2*, the ganglia were gently triturated with pipettes of decreasing diameters (1.32mm, 1.0mm, 0.83mm, 0.45mm). After the trituration step, the cell suspension was diluted within the *Enzyme Inhibition Medium* to remove enzyme activity. To maximize the neuron yield from the mixed cell population and the debris, a cell purification step was incorporated using a three-layer Percoll *Gradient*. Cell suspension was layered on Percoll *Gradient* gently and spun at 1700 RPM at 4˚C. Cells collected from the middle layer were plated with an average density of 100 cells/mm$^2$ using 4.7mm diameter cloning cylinders (Sigma). Half of the maintenance medium was reloaded every three or four days to maintain the viability of the cells up to two months. The details of the preparations and the contents of all the solutions and the media used in our experiments are given in S1 File

### Immunocytochemistry

An MEA plate was fixed on the day of experiment (Table 1, Plate 8) with 4% paraformaldehyde (PFA, pH~6.9) and washed gently with phosphate buffer saline (PBS, Sigma). Blocking and permeabilization of the cells were performed using the *Blocking Solution*. Afterwards, a second wash was performed, and preparation was incubated with primary antibodies in the *Dilution Solution*, overnight at 4˚C. Utilized primary antibodies were chicken Anti β-III Tubulin (Abcam) and rabbit Anti-Synaptophysin (Santa Cruz) with 1:200 and 1:50 dilutions, respectively. After that, the primary antibodies were washed out with PBS and secondary antibodies were added and incubated for three hours at room temperature. Secondary antibodies were Alexa Fluor 633 goat anti-chicken Immuno-globulin-G (IgG) and Alexa Fluor 488 goat anti-mouse IgG (Invitrogen) with 1:100 and 1:400 dilutions, respectively. DAPI (Invitrogen) was added to the sample at 1 μg/ml concentration. A final wash with 2:1000 PBS-Tween20 solution was performed in dimmed light and preparation was kept in PBS-Azide with 1:1000 dilution.

### Preparation of MEAs

MEAs with 64 planar microelectrodes etched to 5cmx5cm glass substrates were purchased from *The Center for Network Neuroscience* of the *University of North Texas*. We removed the

**Table 1. Basic interaction profiles.**

|  | Plate 1 | Plate 2 | Plate 3 | Plate 4 | Plate 5 | Plate 6 | Plate 7 | Plate 8 |
|---|---|---|---|---|---|---|---|---|
| **A** | 52 | 43 | 54 | 74 | 50 | 53 | 49 | 62 |
| **B** | 33 | 27 | 10 | 51 | 14 | 10 | 13 | 23 |
| **C***  | 5 | 4 | 3 | 5 | 6 | 5 | 4 | 5 |
| **D***  | 3 | 3 | 2 | 4 | 3 | 2 | 2 | 3 |
| **E ($\mu m$)** | 1000 | 860 | 730 | 800 | 920 | 900 | 600 | 700 |

A: Total number of neuron bodies on active electrode area ($1mm^2$).

B: Total number of candidate neuron bodies in ε-neighborhood.

C: Number of primary neurons (directly stimulated neurons).

D: Number of primary neurons that excite secondary neurons.

E: Maximal distance between primary and secondary neurons.

* Only significant values (p<0.05) included, determined using Eqs 1 to 4.

glass bottoms of standard 35mm Petri dishes and adhered the remaining polystyrene frames to the MEAs using a medical adhesive (Hollister- 7730). The new MEA dishes were sterilized with 3% bleach and 70% ethanol and placed under UV light for two hours in a laminar flow hood. Afterwards, they were coated with 0.1% polyethyleneimine (PEI, Sigma) solution prepared in 0.1M borate buffer for two hours and rinsed off thoroughly with sterile de-ionized water. The dishes were then coated with $40ng/mm^2$ Laminin (Sigma) diluted in double-distilled water ($ddH_2O$) and kept at 37˚C incubation overnight.

## Local electrical stimulation

The electrical stimulation hardware consisted of an MEA interfacing headstage, a digital to analog conversion board, a router circuitry, and a controlling PC (Lenovo). Plexon MHP64 headstage was used for interfacing MEAs which was designed for the 64 channel MEA layouts. A custom designed router circuitry was employed for directing analog stimulation signals to selected channel or channels from 64 alternatives. Digital stimulation signals were converted to analog voltages using National Instruments (NI) 6001 board, at a sampling rate of 5 kHz. Digital to analog conversion and router circuitry were controlled with a custom software written in MATLAB (Mathworks Inc.) incorporating Data Acquisition Toolbox (Mathworks Inc.) and utilizing the NI drivers and libraries. All stimulation signals were voltage controlled biphasic pulses and the stimulation parameters were adapted from previous studies [17–19]. The pulse durations and the amplitudes that elicit stable and reproducible responses were empirically adjusted in the ranges of 100–500 μs and 1–3 V, respectively.

## Experimental procedure

To observe individual calcium responses clearly, the stimulation period was selected as six seconds, similar to previous studies [19, 20]. Within six second periods, stimulation was applied as dual-pulses in 2 Hz. After the candidate electrodes were determined, the scanning stimulation pulses were applied to each candidate electrode in sequence. In this way, the regions that respond to stimulation were selected. A typical stimulation strategy that was used in the experiments is shown in Fig 1A. The MEA headstage was placed on a custom-made heating unit, similar to a previously reported interface on an inverted confocal microscope stage (Carl Zeiss, Cell Observer) [21]. Time lapse images were acquired simultaneously while applying local electrical stimulation through selected electrodes. For imaging, a 10X Plan Apochromat objective and a 488nm excitation wavelength laser were used along with a 500–550 nm emission filter.

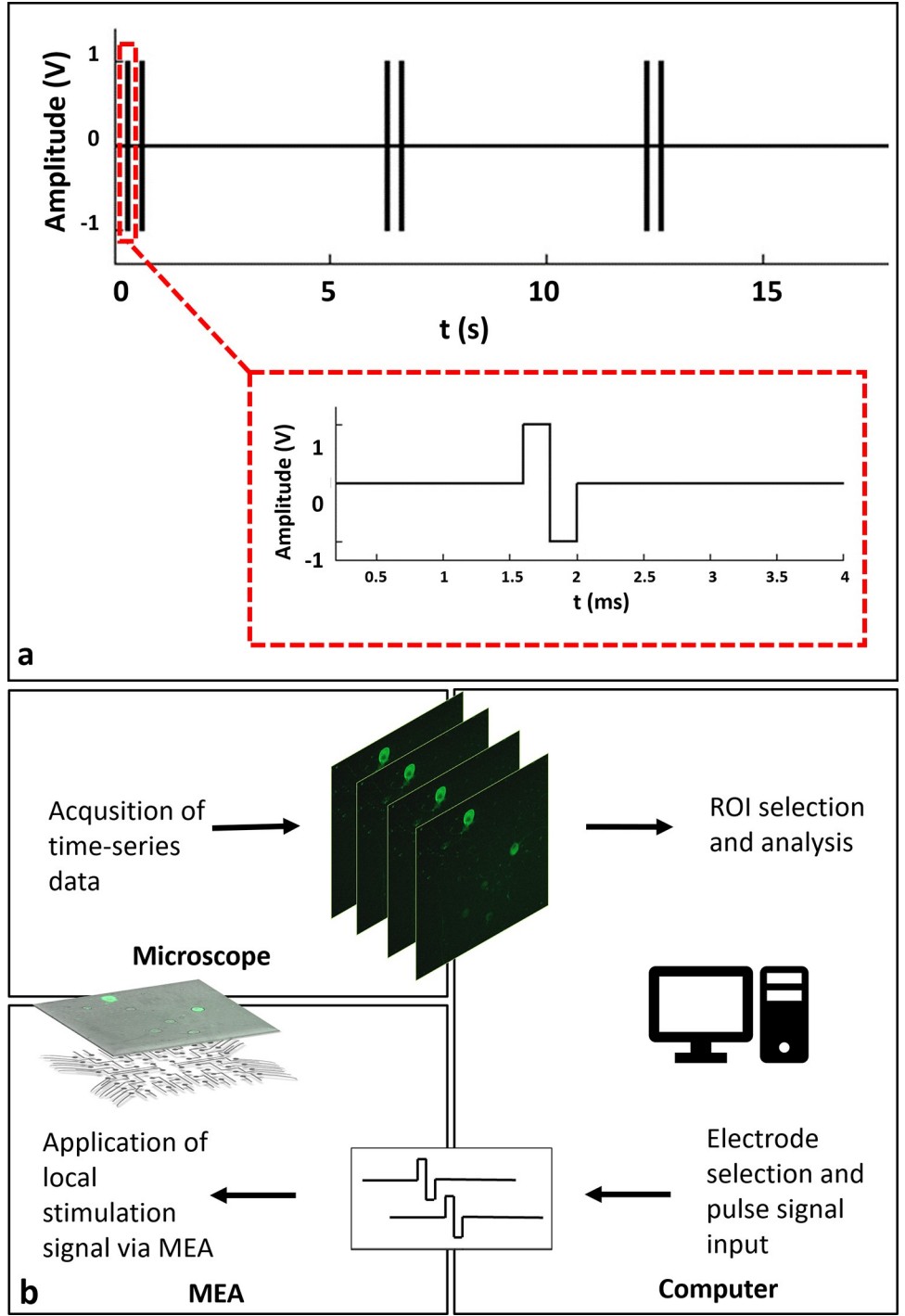

**Fig 1.** a: A typical stimulation strategy. A stimulation consists of two biphasic pulses of $400\mu s$ duration, repeated in 0.5 s. Stimulation is repeated within six second periods. b: Schematic representation of the experimental procedure. Time-lapse imaging and electrical stimulation of GCaMP6s expressing neurons using a MEA headstage placed on an inverted fluorescent microscope. Acquisition of calcium responses and live monitoring. Offline event detection, analysis and rendering of analog voltage signals.

Image acquisition was performed at an average rate of three frames per second which sufficiently captured GCaMP6s dynamics [19]. A schematic diagram of the experiment loop can be seen in Fig 1B. All experiments were carried out with identical stimulation set up in "default (non-blocked)" and "blocked" conditions where NMDA, AMPA and Kainate receptor blockers were applied to the cultures.

## Image analysis and statistics

Neuronal cell bodies were selected as regions of interests (ROIs) manually using Fiji software [22]. A selected set of ROIs and the corresponding stimulation electrode was assigned as an individual experiment. Each pixel that a region encloses, was assumed to have a uniformly distributed variation in light-intensity and calcium activity of each neuronal unit was calculated by averaging. Then the relative change of light intensities, $\Delta F/F(t)$, were computed for each ROI after subtracting the background [23]. In order to eliminate imaging noises and to remove the out-of-range frequency components, a moving-average filter was applied to each time-series [24].

To evaluate the similarity between the responses of the selected ROIs, cross-correlation analysis was applied to each pair of time-series. To obtain a consequent output, a cross-correlation measure for any two time-series, the cross-covariance was computed using (1) and (2),

$$\sigma_{xy}(T) = \frac{1}{N-1}\sum_{t=1}^{N}(x_{t-T} - \mu_x)(y_t - \mu_y)$$

(1)

$$r_{xy}(T) = \frac{\sigma_{xy}(T)}{\sqrt{\sigma_{xx}(O)\sigma_{yy}(O)}},$$

(2)

where, $N$ is the length and, $\mu_x$ and $\mu_y$ are mean values of time series $x$ and $y$, respectively.

The maximum values of the cross-correlation signals were determined, allowing a maximal lag of a stimulation period. Then using lag-compensated cross-correlation results were further processed to find Pearson correlation coefficients with Eq (3),

$$c_{xy} = \frac{N(\sum xy) - (\sum x)(\sum y)}{\sqrt{[N\sum x^2 - (\sum x)^2][N\sum y^2 - (\sum y)^2]}}$$

(3)

The correlation coefficients above a selected threshold was used in connectivity analyses, and their significance was determined using $p$-value statistics calculated using (4).

$$p_{xy} = c_{xy}\sqrt{\frac{N-2}{1-c_{xy}^2}}$$

(4)

To group the calcium responses hierarchically, city-block distance and complete link clustering algorithms were employed. In addition, a phase synchronization index, *Mean Phase Coherence* (MPC), was employed for determining phase coupling strength of the correlated calcium responses [25]. To compute the MPC values, instantaneous phase differences of the time-series were utilized as follows,

$$MPC_{xy} = |\frac{1}{N}\sum_{j=0}^{N-1} e^{i\varphi_{x,y}(j\Delta t)}|$$

(5)

## Results

Healthy populations of adult DRG cells were obtained with a high viability ratio of 90–95%. The cells firmly attached on the MEA surfaces due to the optimized coating protocol. Neurite elongation was observed in the first hour of plating. Plated cell populations included 60–65% neurons, the remaining were glial cells. Fig 2 shows phase contrast (a1) and fluorescent images (a2) of a typical DRG culture grown on an MEA dish. A high-throughput expression of GCaMP6s can be seen on glutamatergic neurons at three days in vitro (DIV3). All the experiments were conducted between DIV2-11. Glutamate receptor antagonists were used for examining postsynaptic connections between the DRG neurons [26]. In the presence of NMDA, AMPA and Kainate receptor blockers, synaptic communication ceased completely. Induced calcium activity patterns changed dramatically when the glutamatergic synapses were blocked. The results were verified with post-control data, obtained after the wash out of the chemical antagonists with fresh media. Experimental steps are outlined in Fig 3.

*Active MEA areas*, approximately a total of 1mm$^2$ region, were monitored with a 10X objective and the total number of neuronal bodies were counted (Table 1, row A). *Primary Neurons* were defined as the neuron bodies which were directly stimulated by an electrode. A circle of radius $\varepsilon$ around each selected electrode, called a $\varepsilon$-*neighborhood*, was determined to be the area where primary neurons reside. *Secondary Neurons* were defined as the neurons which were excited by the primary neurons. They were observed throughout the active MEA area outside the $\varepsilon$-neighborhood, called the *complementary neighborhood*. An illustrative diagram for neighborhood definitions can be found in Fig 2B. In the reported experiments, we selected the neighborhood parameter as, $\varepsilon = 50\ \mu$m. Table 1 summarizes the basic interaction profiles of each MEA plate. Out of a total of 437 neurons in the active areas of these eight MEA plates (Table 1, row A), 181 were in the $\varepsilon$-neighborhood (Table 1, row B). These neurons in the $\varepsilon$-neighborhood were defined as *candidates* for primary neurons. To determine the primary neurons, we stimulated the candidates via selected electrodes in the $\varepsilon$-neighborhood. A total of 37 candidate neurons responded to the stimulation ($p<0.05$) and they were labeled as the primary neurons (Table 1, row C). Out of these 37 primary neurons, 22 were found to excite at least one other neuron (60% ± 15.7 with a confidence interval of 95%, $p<0.05$) (Table 1, row D). The neurons that were excited by the primary neurons were the secondary neurons. The last row of Table 1 shows the maximal distances between primary and secondary neurons.

Two distinct response profiles were observed depending on the presence and the absence of a contact between the stimulation electrode and the primary neuron. If a direct contact was

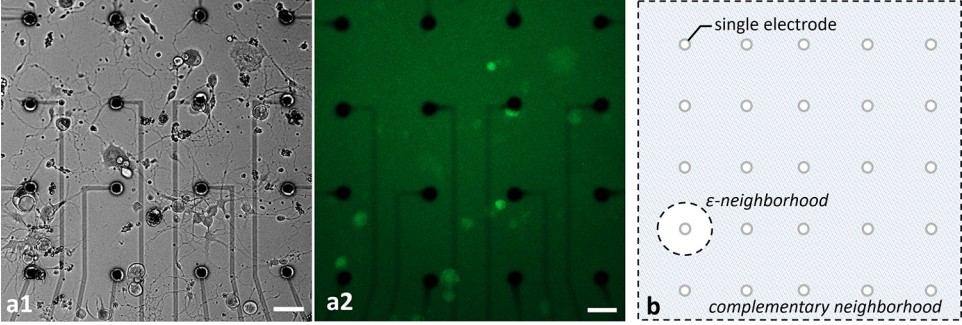

**Fig 2.** Phase contrast (a1) and fluorescent (a2) images of typical cultures of DRG neurons. The cultures were grown on MEA plates with an average density of 100 cells/mm$^2$. Fluorescent image shows the GCaMP6s expression of transgenic glutamatergic neurons at DIV3. Scale bar: 50 $\mu$m at 10x magnification. The illustration of $\varepsilon$- and complementary neighborhood definitions (b).

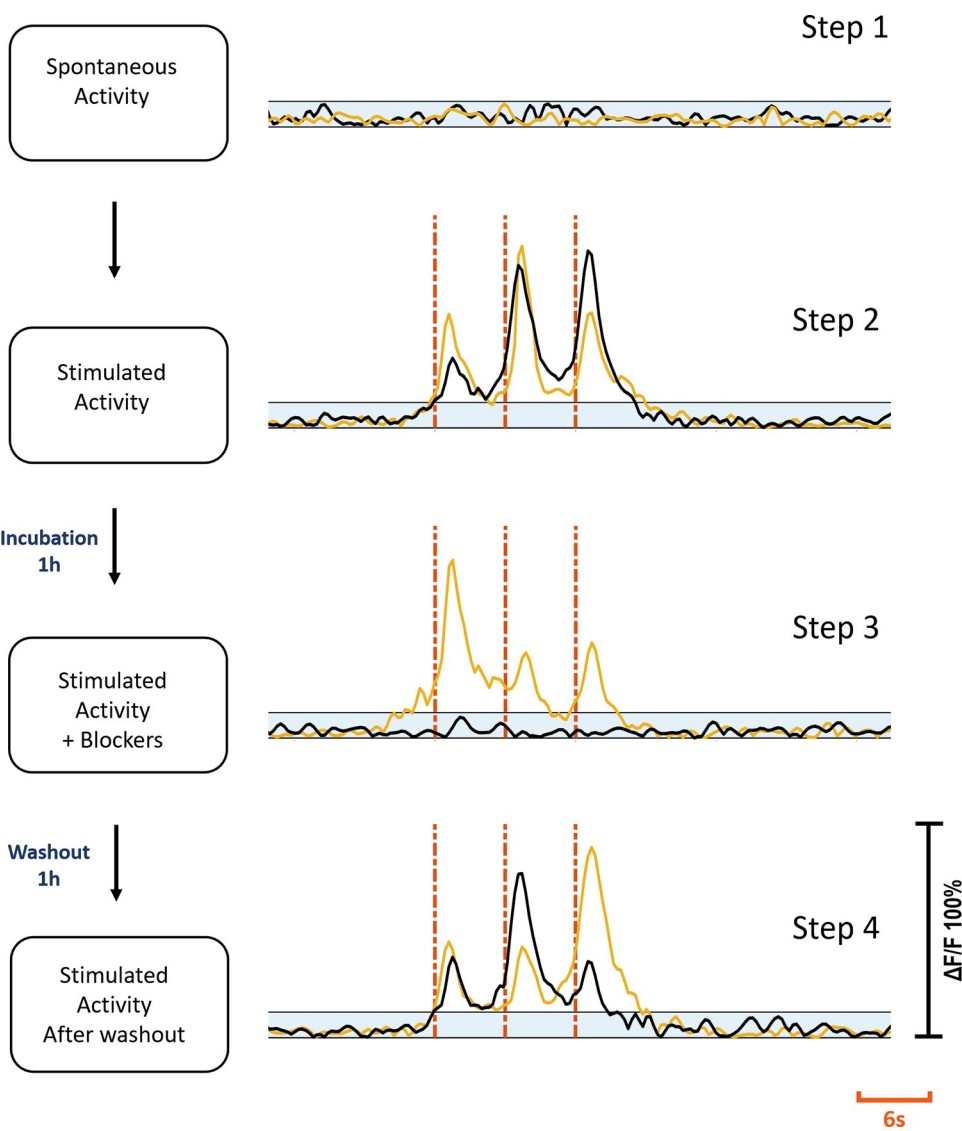

**Fig 3. Experimental steps.** Step one: Spontaneous activity recording prior to stimulation. Step two: electrical stimulation applied. Step three: Stimulation repeated after cultures were treated with blockers for 1h. Step four: Stimulation repeated after washout and 1h incubation. Yellow and black traces show the same $c^+$ and $c^-$ primary neuron's activities at each step respectively. Blue shaded areas show subthreshold. Red dashes show stimulation instants. Each stimulation trace corresponds to a 2 Hz dual-biphasic pulses repeated in 6s periods. Vertical axis shows normalized ΔF/F.

present, it was defined as the $c^+$ *primary neuron* and if there were no direct contact it was defined as the $c^-$ *primary neuron*. Out of 37 primary neurons, 21 (57%) were of $c^+$ type, and 16 (43%) were of $c^-$ type. For the $c^+$ primary neurons, synaptic blockers did not impede the evoked responses, however for the $c^-$ primary neurons, the evoked responses were suppressed with blocker application. Fig 3 shows the characteristic response profiles obtained from $c^+$ and $c^-$ type primary neurons at each experimental step. The responses of each $c^+$ and $c^-$ primary neuron (n = 13 for $c^+$, n = 10 for $c^-$) recorded in default and blocked conditions can be seen in Fig 4.

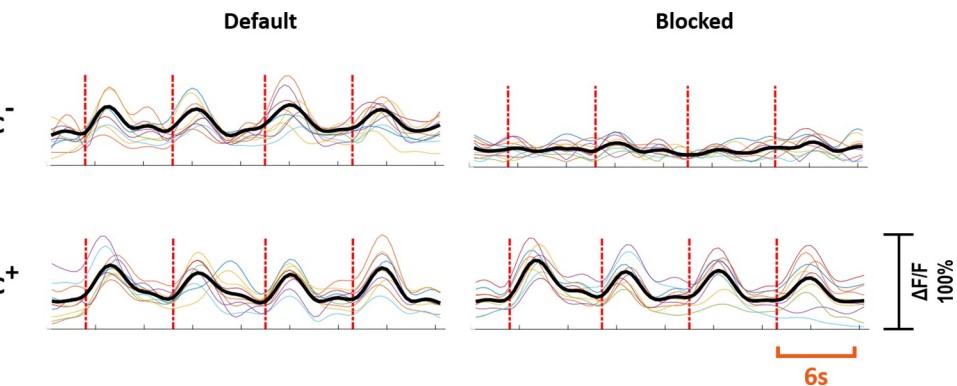

**Fig 4. Response profiles of c⁺ and c⁻ neurons.** Individual experiments are shown in color and the averaged response is shown in black. The first row shows the responses of c⁻ neurons and the second row shows the responses of c⁺ neurons. The left and right column shows responses to stimulation in default and blocked cases, respectively. Red dashed lines show stimulation instants. Stimulation period is six seconds. Vertical axis shows normalized ΔF/F.

## Stimulated neuronal responses activate other neurons via glutamatergic connections

Interaction profiles between primary and secondary neurons are illustrated in Fig 5, for both c⁻ (a) and c⁺ type primary neurons (b and c). In (Fig 5A), the c⁻ primary neuron (ROI 1) excites two secondary neurons (ROIs 2 and 3). Locations of these neurons and the stimulation electrode can be seen from the fluorescent image presented in Fig 5A1. Recorded $Ca^{2+}$ responses of these primary and secondary neurons are shown in Fig 5A2. Correlation analysis confirmed that the secondary neurons were excited by the c⁻ primary neuron ($p < 0.05$). In the blocked condition, no $Ca^{2+}$ activity was observed.

Although the induced $Ca^{2+}$ activity persisted under the glutamatergic inhibition for c⁺ primary neurons, excitability of the secondary neurons discontinued. Secondary neurons exhibited two common excitation profiles depending on whether they were silent or spontaneously active before stimulation. These two profiles were encountered in equal frequencies 30% (both 7 out of 22) all throughout the experiments. In Fig 5B, the c⁺ primary neuron was observed to excite a previously silent secondary neuron. In the blocked case this excitation was not observed as shown via correlation analysis ($p < 0.05$). A video of this experiment is provided in S1A and S1B Video. Fig 5C shows an example for a secondary neuron which was spontaneously active before stimulation. Spontaneous activity of the secondary neuron was modulated with the induced $Ca^{2+}$ activity of the primary neuron. Excitation of the secondary neuron continued through the stimulation and the in-phase modulation vanished in the blocked case ($p < 0.05$).

## Higher order synaptic interactions

Increasing the confluency of the cultures produced rather complex interaction patterns. A complex interaction scheme is presented in Fig 6, where a c⁻ primary neuron excites eight secondary neurons. In the blocked repetition of the experiment, interaction between the primary and the secondary neurons disappeared ($p < 0.05$). Recorded $Ca^{2+}$ activities of these neurons are shown in Fig 6A, as $\Delta F/F(t)$ time series. A stimulation strategy involving three amplitude steps (2V-3V-2V) was employed to investigate the effect of the successive stimulation on the excitability. Fig 6B shows the acquired responses to each stimulus for three voltage steps, indicated as (i, ii and iii). A summary of the averaged and normalized responses is presented in Fig

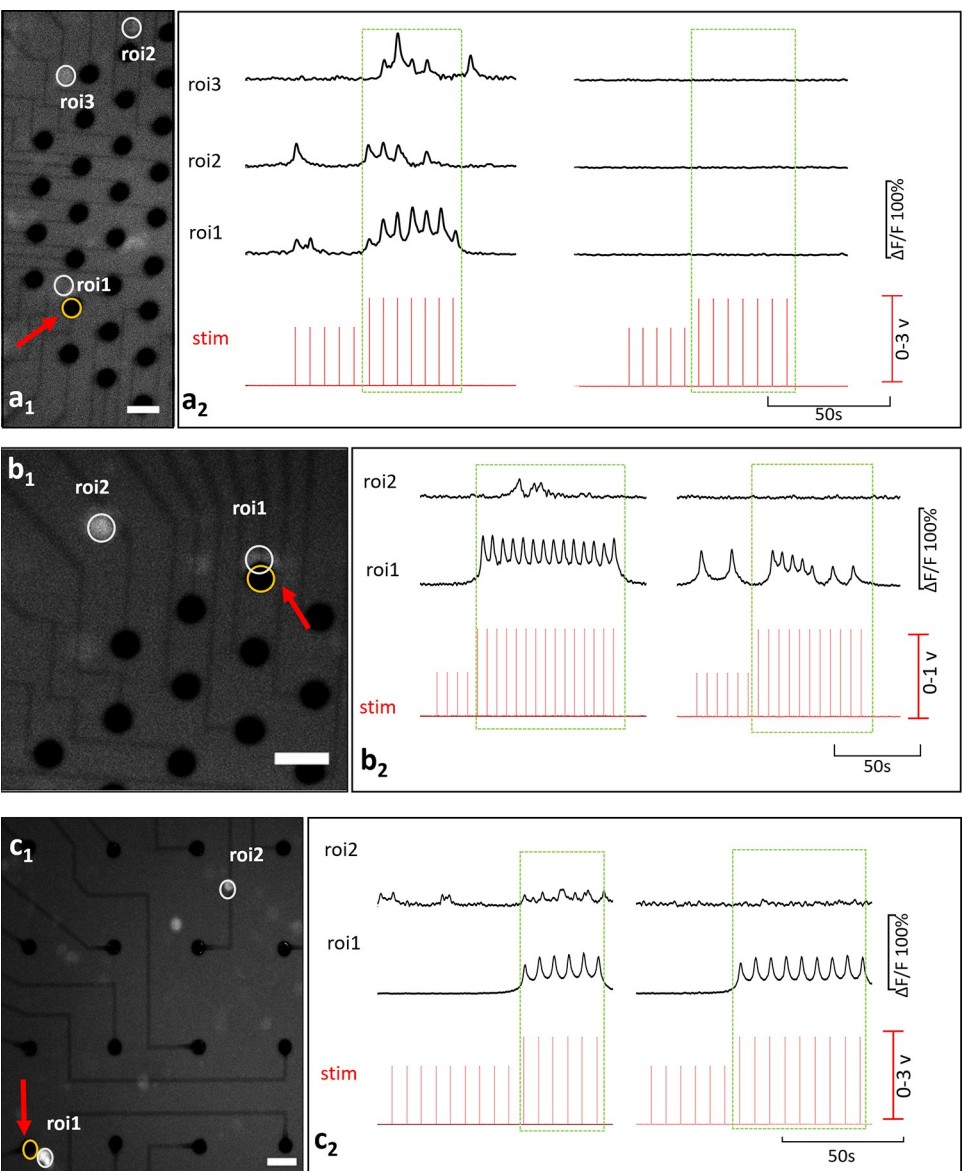

**Fig 5. Basic interactions.** (a) is an example for a c⁻ primary neuron and (b) and (c) are for c⁺ primary neurons. Secondary neurons in (b) and (c) represent previously silent and previously spontaneously active types respectively. Fluorescent images show the locations of the stimulation electrode and the neuron bodies in a1, b1, and c1 (Scale bar: 50 μm, magnification: 10X). Stimulation instants and the corresponding time-series of calcium activity recorded in absence (left) and presence (right) of glutamatergic antagonists, a2, b2, c2.

6C. Significant increase in the excitability of ROIs 1, 4, 5 and 9 was observed ($p < 0.05$) and shown with asterisks. Furthermore, we investigated connectivity, using cross correlation and MPC analyses, separately for the three stimulation periods (i, ii, iii). Significant connections are illustrated as hierarchical trees, shown in Fig 6D ($p < 0.05$). In these three stimulation periods, network structures varied through lower orders of hierarchical connections. Connected pairs that were common in these stimulation periods were then investigated in terms of the connectivity strengths. Significant alterations in the functional connections are summarized in Table 2 and Fig 6E ($p < 0.05$).

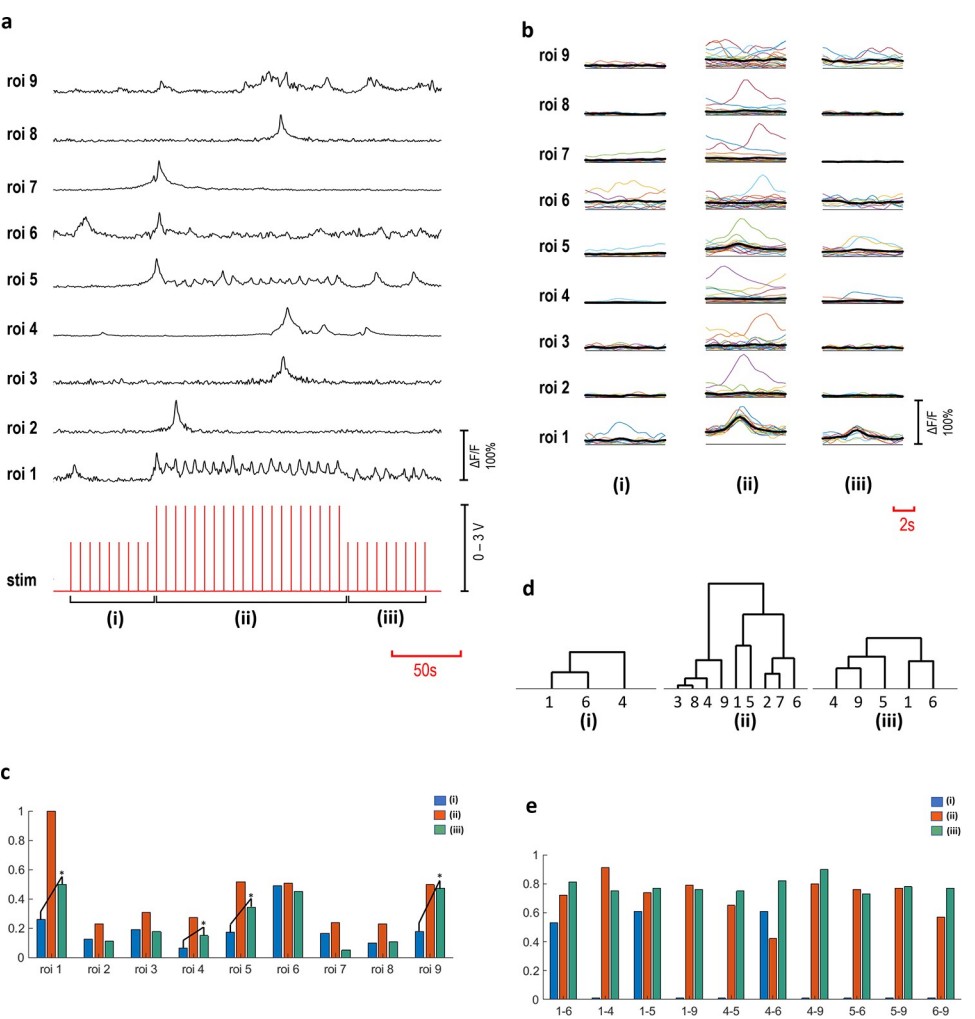

**Fig 6. Multi-layered interactions.** Time-series of ROIs 1–9, ROI 1 is a c⁻ primary neuron and ROIs 2–9 are secondary neurons (a). The stimulation voltage applied in three steps (2V-3V-2V) indicated as (i, ii, iii). (b) shows the averaged responses to individual stimuli detected in three voltage steps (n = 10 for (i) and (iii) steps, n = 20 for (ii) part). Response amplitude alterations are shown in normalized averaged values (c). Response amplitudes of ROIs 1, 4, 5 and 9 increased significantly from step (i) to (iii) (p<0.05). Hierarchical groups are shown, based on significant correlations (p<0.05) (d) and the connectivity strength values calculated with significant MPC scores (p<0.05), (e) obtained from connectivity analyses.

## Structural verification of synapses

To examine synaptic formations on MEA plates, immunostaining was performed (Table 1, Plate 8). Synaptophysin staining was observed among almost all the tubulin positive neurons, prominently concentrated in three distinct regions. These three regions were axon terminals, soma-axon contact sites and axon-axon intersection points. These regions may correspond to synapse types commonly described as, axo-extracellular, axo-synaptic, axo-somatic and axo-axonic (Fig 7). Fig 7.1 and 7.2 show a soma-axon connection and an axon terminal, respectively. The axon-axon intersection sites were observed to be widespread. The a, b, c and d labels show tubulin in red, synaptophysin in green, DAPI and brightfield channels respectively. A 3D rendering of ICC image can be seen in S2 Video.

**Table 2. Multi-layered interaction profiles.**

|  | i | ii | iii |
|---|---|---|---|
| ROIs 1 – 6 | 0.61 | 0.74 | 0.77 |
| ROIs 1 – 4 | 0.53 | 0.72 | 0.81 |
| ROIs 1 – 5 | - | 0.91 | 0.75 |
| ROIs 1 – 9 | - | 0.79 | 0.76 |
| ROIs 4 – 5 | - | 0.65 | 0.75 |
| ROIs 4 – 6 | 0.61 | 0.42 | 0.82 |
| ROIs 4 – 9 | - | 0.80 | 0.90 |
| ROIs 5 – 6 | - | 0.76 | 0.73 |
| ROIs 5 – 9 | - | 0.77 | 0.78 |
| ROIs 6 – 9 | - | 0.57 | 0.77 |

* Only significant MPC scores (p<0.05) included, determined using Eqs 4 and 5.

## Discussion

In this study, we investigated cultured DRG neurons in terms of their spontaneous and induced electrical activity patterns and asked whether they develop networks with each other. For this purpose, we developed an experimental platform combining fluorescent imaging and local electrical stimulation of GCaMP6s expressing sensory neurons from adult mice. We have shown that correlated activity in cultured DRG neurons originate from network formation. Network events were successfully addressed to glutamatergic synapses which were detected in the axonal terminals, axon-soma junctions and axon-axon intersection sites in almost all neurons. Studies using a multi-modal experimental approach to understand interaction profiles of DRG neurons in-vitro are rare and to our knowledge, no study on DRG networks has not been reported previously.

The culture protocol for dissociated DRGs from adult mice with a serum-free protocol provided high viability rates and adequate conditions for growth and development of neurons. Also, the gradient-based cell sorting process resulted in seeding of a low percentage of glial cells with a high neuronal population compared to previous studies. Avoiding the need for anti-mitotic agents by using an optimized culture medium for neurons [27], we reduced the stress and improved the life span of the cultures. To evoke spontaneous electrical and calcium activity in DRG cultures, use of growth factors and cytokines such as NGF, BDNF or GDNF is a common practice [11, 12]. However, synthetic sensitization of the neurons may bring unexpected interactions [28, 29]. Cheng et al. found out that elevation in NGF levels builds up synapse-like structures between sprouted neurites, resulting in mechanical hypersensitivity of healthy neurons [30]. Accordingly, to preserve physiological activity profiles as much as possible, we did not resort to inflammation models for sensitization.

Electrical and optical techniques together were used by Wainger et al. for tracking the activity of nociceptor neurons re-programmed from fibroblasts in order to develop a model for pain research [31]. Following that, Enright applied simultaneous Fluo-8 imaging with MEA-recording for investigating primary human DRG neurons exposed to chemical stimulants [32]. Fluo-8 has a better temporal resolution compared to the GECIs, however they are non-selective, run for limited durations of time and require strict dye-loading protocols [33]. In our approach, we used a targeted GECI to track only the glutamatergic DRG neurons, since glutamate is the presumed neurotransmitter between the DRG and the spinal cord [34]. Vesicular glutamate transporters vGLUTs, found in glutamatergic neurons, can be employed for targeting and identification of DRG neurons, particularly the vGLUT-2 subtype which is

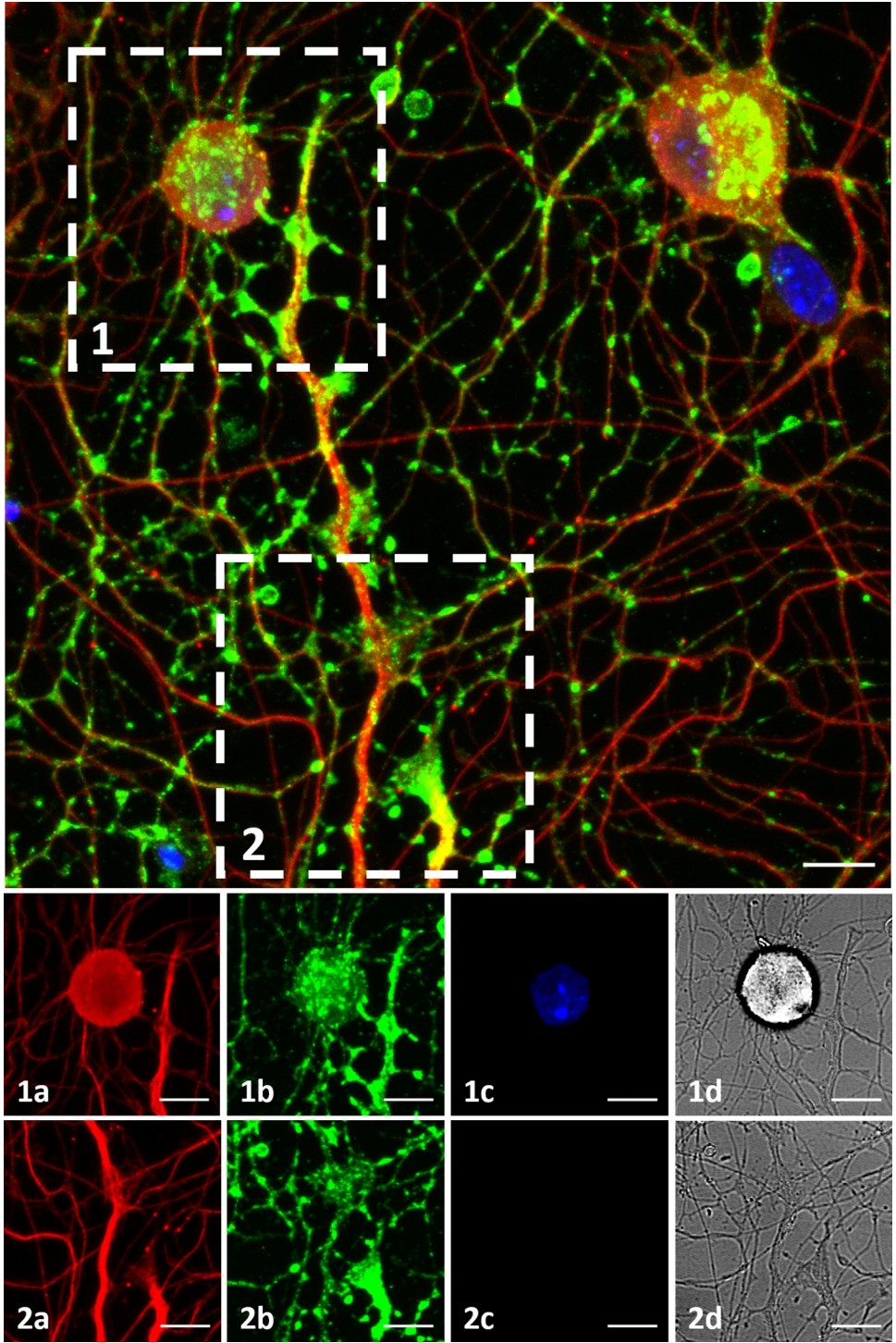

**Fig 7. Immunocytochemistry.** DRG culture on a MEA plate fixed on DIV3. Top: Figure shows maximum intensity projection of 14 z-stack images acquired in various depths, merging 3 channels. β-III Tubulin is shown in red, Synaptophysin is shown in green and DAPI is shown in blue. Bottom: (1) and (2) show a soma-axon connection and an axon terminal in detail. The axon-axon intersection sites are observed widespread. The a, b, c and d labels show tubulin, synaptophysin, DAPI and brightfield channels. Objective: 40X, scale bar:10μm.

found broadly and expressed more in medium to small, nociceptive neurons [26, 35]. The animal model we used ensures correct identification of GCaMP6s expressing vGLUT-2 positive DRG neurons.

The stimulation, protocols used in previous works were usually limited to chemical applications [12, 31, 32] and modality-specific stimuli like heat and cold [11] or mechanical stress [20]. In our protocol, we employed local electrical stimulation with an almost single-cell precision and bypassed the unselective applications of modality-specific assays. As a result, by selective application of stimulus trains, we intended to imitate the encoded sensory signals which may originate from any modality.

In a previous related study, Newberry et al. developed spontaneously active DRG cultures grown on MEA plates in order to study sensory neurons within a network context [12]. Afterwards, Black et al. conducted experiments with DRG neurons on multi-well MEA platforms and observed synchronous and correlated activity, which was discussed to be originating from gap junctions but not synaptic connections [11]. By using glutamatergic receptor antagonists, we successfully blocked the interconnected activity and showed the existence of glutamatergic post-synapses in cultured DRG neurons functionally. Subsequently, we investigated the presence of synapse formations structurally using ICC technique. Synapses were effectively stained with the pre-synapse marker synaptophysin. These successive findings, for the first time, confirmed the materialization of complete synapses and formation of synaptic networks in the DRG cultures.

The first part of our extracellular electrophysiology experiments defines the primary neuronal responses to stimulation and, the second part shows the basic interaction profiles between the primary and the secondary neurons. The third part investigates the network properties involving interactions of multiple neurons. Analyses summarized in Fig 6 shows the excitability alterations and thus the decreased thresholds of excitation. In addition, it is found that the number of connected pairs are also increased after repetitive stimulation. The significant changes occur due to repetitive stimulation and these findings suggest an underlying synaptic facilitation mechanism [26].

Causal transitions of the neuronal activity are not covered in this study and exact axonal tracing is left for future studies. Since the connections are essentially axonal, instead of only monitoring the neuron bodies, use of an axonal-GCaMP indicator could provide more information by allowing imaging of axonal calcium transitions [36]. Since dissociated cell culture models lack organizational structure and deviate from in vivo conditions, we recommend additional experiments involving explant or slice cultures which would be more confluent with denser interactions.

We developed a versatile setup to study the network behavior of adult DRG neurons in vitro. This setup combines MEAs for stimulation and genetically encoded calcium indicator (GECI) based monitoring. The sensitivity achieved by $Ca^{2+}$ imaging allows recording from adult DRG neurons in vitro without resorting to any inflammation model. Evoked responses from cultured DRG neurons through almost single-cell stimulations showed similarities between individual responses and correlation analyses verified statistical relationships between neurons. We demonstrated that this correlation originates from functional synaptic connections using glutamatergic post-synaptic blockers. Applying pre-synaptic marker synaptophysin, we verified the presence of synapses also structurally. Multi-layer network experiments revealed that continuous stimulation increases coupling strength of neurons. Our results suggest a new type of neuron-to-neuron interaction conducted through synaptic connections in cultured DRG neurons in which a stimulated neuron either modulates spontaneous activity of other neurons or activates previously quiet neurons.

Somata of sensory neurons do not form any synapses with each other inside the DRG in-vivo [2, 37–39]. However, we have shown synapse formation between DRG neurons in vitro. In dissociated culture model, DRG neurons are released from ganglion structure and, connective and glial tissue layers are removed. This alteration can be a cue for synapse formation. In addition, the dissociation procedure itself may act as an injury model which is known to reprogram DRG neurons and temporarily alter cell identity [40, 41]. Functional and structural synapse formation and network development potential of sensory neurons may explain neuron-to-neuron interactions in a new scope [3]. These findings may shed new light on various disorders such as neuropathies, fibromyalgia, small fiber neuropathy, immune-mediated hyperalgesia, and other pain syndromes of peripheral nervous system [42, 43].

## Supporting information

**S1 File. Solutions and media.**
(DOCX)

**S1 Video. Basic interactions.**
(ZIP)

**S2 Video. Immunocytochemistry.**
(MP4)

## Author Contributions

**Conceptualization:** F. Kemal Bayat, Halil Özcan Gülçür, Albert Güveniş.

**Data curation:** F. Kemal Bayat, Betul Polat Budak, Esra Nur Yiğit.

**Formal analysis:** F. Kemal Bayat, Betul Polat Budak, Esra Nur Yiğit.

**Funding acquisition:** Albert Güveniş.

**Investigation:** F. Kemal Bayat, Betul Polat Budak.

**Methodology:** F. Kemal Bayat, Betul Polat Budak, Esra Nur Yiğit.

**Project administration:** F. Kemal Bayat, Albert Güveniş.

**Resources:** F. Kemal Bayat, Esra Nur Yiğit, Halil Özcan Gülçür, Albert Güveniş.

**Software:** Halil Özcan Gülçür.

**Supervision:** Gürkan Öztürk, Halil Özcan Gülçür, Albert Güveniş.

**Visualization:** F. Kemal Bayat, Betul Polat Budak.

**Writing – original draft:** F. Kemal Bayat, Betul Polat Budak.

**Writing – review & editing:** F. Kemal Bayat, Betul Polat Budak, Esra Nur Yiğit, Gürkan Öztürk, Halil Özcan Gülçür, Albert Güveniş.

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
