## [Decision Letter · Decision Letter 0]

9 Dec 2020

PONE-D-20-30530

Adult mouse dorsal root ganglia neurons form aberrant glutamatergic connections in dissociated cultures

PLOS ONE

Dear Dr. Gülçür,

I'm really sorry for the delay. Thank you for submitting your manuscript to PLOS ONE. After careful consideration, we feel that it has merit but does not fully meet PLOS ONE’s publication criteria as it currently stands. Therefore, we invite you to submit a revised version of the manuscript that addresses the points raised during the review process.

I agree with both reviewers that your study needs just some minor changes. From my point of view I'd like to add one point. In line with one of the reviewers -  Line 362: “Although it is widely accepted that peripheral neurons do not make synapses between each other…”.

Actually, I'm not so sure whether this is true, because there is some important literature already available (e.g. https://pubmed.ncbi.nlm.nih.gov/2862229/). I'd like to suggest to discuss some of the available findings in context of your study. I think it is worth to discuss this in a complete paragraph.

We look forward to receiving your revised manuscript.

Kind regards,

Robert Blum

Academic Editor

PLOS ONE

Journal Requirements:

2. Please amend either the title on the online submission form (via Edit Submission) or the title in the manuscript so that they are identical.

Reviewers' comments:

Reviewer's Responses to Questions

**Comments to the Author**

1. Is the manuscript technically sound, and do the data support the conclusions?

Reviewer #1: Yes

Reviewer #2: Yes

2. Has the statistical analysis been performed appropriately and rigorously? 

Reviewer #1: Yes

Reviewer #2: Yes

3. Have the authors made all data underlying the findings in their manuscript fully available?

Reviewer #1: Yes

Reviewer #2: Yes

4. Is the manuscript presented in an intelligible fashion and written in standard English?

Reviewer #1: Yes

Reviewer #2: Yes

5. Review Comments to the Author

Reviewer #1: Primary sensory neurons detect noxious and non-noxious signals in the body periphery and forward the information to the CNS via secondary neurons located in the spinal cord. Cultures of dissociated primary sensory neurons obtained from dorsal root ganglia (DRG) of rodents are widely used to model sensory neuron-associated disease conditions and to study the processing of sensory information on molecular and cellular levels.

In their manuscript “Adult mouse DRG neurons form aberrant glutamatergic connections in dissociated cultures” Bayat et al. analyze dissociated low-density DRG cultures by means of calcium imaging and extracellular electrophysiology to infer about functional neuron-to-neuron interactions. In their study they used DRG neurons from mice expressing the genetically encoded Ca2+ indicator GCamp6s specifically in the glutamatergic neuronal subpopulation. They stimulated individual GCamp6s-positive neurons with biphasic voltage pulses and recorded the calcium responses from the stimulated as well as from distant GCamp6s-positive neurons. Using this approach, they showed that a significant proportion of stimulated neurons can trigger secondary responses in distant neurons demonstrating direct functional coupling between them. Interestingly, functional coupling vanished upon application of inhibitor solution consisting of AP5 and CNQX suggesting that the observed neuron-to-neuron communication was mediated by functional glutamatergic synapses formed between individual neurons. They further confirmed in immunocytochemistry experiments a widespread expression of synaptophysin at axon-axon intersections, axon terminals and soma-axon contact points.

The identification of active functional glutamatergic synapses between individual cultured DRG neurons is an important information for all working with this model system. The presented dataset is original and interesting to the broad readership of PLOS ONE. However, there are a couple of points that require the attention of the authors and need to be clarified:

Minor points:

1) Methods: The stimulation procedure should be explained more precisely. The authors just show a voltage stimulation protocol in Fig.1 and state that a 6-second stimulation period was used (line 132) and that all pulses were biphasic (line 128 onwards). Fig. 1 shows a two-pulse stimulation paradigm; was a double-pulse stimulation used in all experiments? What was the duration between the two individual pulses? The red traces in Fig.5 and Fig.6a suggest that monophasic pulses were used. Is this correct or are these traces just stimulation indicators?

2) The term “DIV” is not specified (eg. Line183 and others). Is this the abbreviation for “days in vitro”?

3) The authors state that all experiments were performed with young cultures between DIV2-12 (line 184) but also note that neurons were cultivated for up to two months (line 96). Did they use also older cells beyond age DIV12 or what was the reason to restrict experiments to young cultures and/or to cultivate cells for up to two months?

4) Most references are provided as numbers in paratheses, sometimes author names are used (e.g. line 331 and others). I suggest sticking to one rule.

5) Results: It is stated that 90-95% of cells were viable, 60-65% of plated cells were neurons. It is not clear how these numbers were obtained.

6) Fig.4: I guess c- and c+ traces were obtained from the same cells before and after application of blockers? Maybe this can be stated more clearly in the text or the associated figure legend. How was this experiment performed; first control measurements followed by measurements with blockers present or vice versa? The text (line 187) suggests that the block is reversable. The authors should consider a panel showing the signals of one or more cells under control conditions, in presence of blockers and after washout. How long was the wash-out period? Why is there no scale bar for deltaF/F signals? Since the traces are responses to timed stimulations, I suggest indicating the stimulation time points.

7) I wonder why the rising and falling phases of the traces in Fig.4 appear almost identical. To my knowledge GCamp6s signals obtained with neurons are characterized by a rather fast rising phase followed by a slower decay. This feature can, at least partially, seen in Figs.5 and 6. Is this difference a consequence of data processing, i.e. the smoothing with a moving average? How many datapoints were averaged to smooth data?

8) Table 2 and Fig.6f are identical. Is there a reason for that?

9) The authors used calcium signals as surrogate for electrical activity of individual neurons. Did the authors also record extracellular action potentials from active neurons with their MEA system? Although beyond the scope of this study, it would be a valuable information and could further support their findings.

10) Line 362: “Although it is widely accepted that peripheral neurons do not make synapses between each other…” Can the authors provide a refence for this statement?

11) Line 363: “Functional and structural synapse formation and network development potential of the cultured sensory neurons may shed new light on the unexplained interactions observed in various disorders of peripheral nervous system.” Which “unexplained interactions” are meant by the authors?

Reviewer #2: This is a well-written manuscript. My only major concern is the quality and labeling of the figures. The resolution is too low and labels such as c+ and c- rather than words make the story harder than necessary to follow. Other than that, my comments are entirely minor in nature:

1. Introduction, line 41: edit "wanted to prove that" to "wanted to examine whether or not"

2. Introduction, line 59: edit "transactions" to "transients"

6. PLOS authors have the option to publish the peer review history of their article (what does this mean?). If published, this will include your full peer review and any attached files.

Reviewer #1: No

Reviewer #2: No

---

## [Author Response · Author response to Decision Letter 0]

23 Jan 2021

Dear Robert Blum

Academic Editor

PLOS ONE,

I would like to thank you and the reviewers for rigorously and meticulously reviewing our manuscript “PONE-D-20-30530 Adult mouse dorsal root ganglia neurons form aberrant glutamatergic connections in dissociated cultures” and making many constructive suggestions. We implemented these valuable suggestions in our revised manuscript carefully and revised the style requirements and applied the required changes. 

Our answers to the issues kindly brought to our attention by yourself and the reviewers are given the attachment “Response to Reviewers”.

Prof. Dr. Halil Özcan Gülçür

Biruni University Department of Biomedical Engineering

Bogazici University Institute of Biomedical Engineering

Response to the Academic Editor,

Comment to the Authors:

In line with one of the reviewers - Line 362: “Although it is widely accepted that peripheral neurons do not make synapses between each other…”.

Actually, I'm not so sure whether this is true, because there is some important literature already available (e.g. https://pubmed.ncbi.nlm.nih.gov/2862229/). I'd like to suggest discussing some of the available findings in context of your study. I think it is worth to discuss this in a complete paragraph.

Answer and Action:

We thank the editor for bringing this issue to our attention. We incorporated the suggested changes in order to clarify the context in the introduction and the discussion sections of the manuscript (p:2 l:36-38; p:15 l:375). 

To avoid any misunderstandings, we included an illustration pointing out to the intended use of our statement: “… peripheral neurons do not make synapses between each other…”. Sensory information is transferred to the spinal cord through a single neuron whose body resides in the DRG, and there are no reported inter-ganglionic synapses in the literature.

Response to Reviewer #1:

Comments to the Authors:

1) Methods: The stimulation procedure should be explained more precisely. The authors just show a voltage stimulation protocol in Fig.1 and state that a 6-second stimulation period was used (line 132) and that all pulses were biphasic (line 128 onwards). Fig. 1 shows a two-pulse stimulation paradigm; was a double-pulse stimulation used in all experiments? 

What was the duration between the two individual pulses? 

The red traces in Fig.5 and Fig.6a suggest that monophasic pulses were used. Is this correct or are these traces just stimulation indicators?

Answers and Actions:

We thank the reviewer for the careful observations and suggestions. The pulses were indeed 2Hz dual-pulses (each pulse with 400µs duration) applied once in every 6 seconds. This paradigm was used in all experiments. In figures 3, 4, 5 and 6, the red traces indicate the stimulation instants. At each instant the same stimulation paradigm was applied as shown in Fig 1. The mentioned changes were implemented in the manuscript and in the figure legends (p:5-l:136, p:6-l:148,149)

Comments to the Authors:

2) The term “DIV” is not specified (eg. Line183 and others). Is this the abbreviation for “days in vitro”?

Answers and Actions:

We included a definition for this term in the revised manuscript. (p:7 l:185)

Comments to the Authors:

3) The authors state that all experiments were performed with young cultures between DIV2-12 (line 184) but also note that neurons were cultivated for up to two months (line 96). Did they use also older cells beyond age DIV12 or what was the reason to restrict experiments to young cultures and/or to cultivate cells for up to two months?

Answers and Actions:

The culture protocol developed by our lab yields cultures viable for more than 2 months. This statement was added to the methods section to let readers know if they would like to replicate.

The experiments were restricted to DIV2-12. As cultures age, glial cells proliferate, and it gets harder to find cell locations within densely populated cultures. Suppressing the proliferation of the glial cells is another option which we do not prefer due to potential harm to neurons.

Comments to the Authors:

4) Most references are provided as numbers in paratheses, sometimes author names are used (e.g. line 331 and others). I suggest sticking to one rule.

Answers and Actions:

We made the appropriate changes as suggested. (p:2 l:49; p:13 l:343,345)

Comments to the Authors:

5) Results: It is stated that 90-95% of cells were viable, 60-65% of plated cells were neurons. It is not clear how these numbers were obtained.

Answers and Actions:

The viability analyses were performed before seeding the cells, via two methods; (1) an automated cell counter (Muse cell analyzer), (2) a manual-counting method using trypan blue staining. For neuron ratios, we performed ICC in early phases of the experiments (Div0-3, unpublished data).

Comments to the Authors:

6) Fig.4: I guess c- and c+ traces were obtained from the same cells before and after application of blockers? Maybe this can be stated more clearly in the text or the associated figure legend. How was this experiment performed; first control measurements followed by measurements with blockers present or vice versa? The text (line 187) suggests that the block is reversable. The authors should consider a panel showing the signals of one or more cells under control conditions, in presence of blockers and after washout. How long was the wash-out period? Why is there no scale bar for deltaF/F signals? Since the traces are responses to timed stimulations, I suggest indicating the stimulation time points.

Answers and Actions:

We thank the reviewer for pointing out the ambiguity in our explanation of the experimental steps. A new figure showing the steps and summarizing characteristic results was added to the revised manuscript (p:8 l:197-202, new Figure 3). In each step, data were obtained from the same cells. 

The order of experimental steps and data obtained after washout are included in the new Figure 3. Wash out and incubation periods are also indicated in the figure along with stimulation traces and scale bars.

Comments to the Authors:

7) I wonder why the rising and falling phases of the traces in Fig.4 appear almost identical. To my knowledge GCamp6s signals obtained with neurons are characterized by a rather fast rising phase followed by a slower decay. This feature can, at least partially, seen in Figs.5 and 6. Is this difference a consequence of data processing, i.e. the smoothing with a moving average? How many datapoints were averaged to smooth data?

Answers and Actions:

We thank the reviewer for mentioning this crucial point. We renewed the figure, decreasing the moving average based smoothing. In the new Figure 4, we used 3 datapoints for averaging the data. We also included the stimulation instants and revised the scale bars.

Comments to the Authors:

8) Table 2 and Fig.6f are identical. Is there a reason for that?

Answers and Actions:

The table(f) in Figure 6 was omitted.

Comments to the Authors:

9) The authors used calcium signals as surrogate for electrical activity of individual neurons. Did the authors also record extracellular action potentials from active neurons with their MEA system? Although beyond the scope of this study, it would be a valuable information and could further support their findings.

Answers and Actions:

We thank the reviewer for this helpful suggestion. The MEA recordings were left out of the scope for this study. We intended to monitor connectivity thoroughly. Electrical recording does not provide the spatial information our experiment requires, since a secondary neuron is very unlikely to be in close vicinity of a recording electrode.

Comments to the Authors:

10) Line 362: “Although it is widely accepted that peripheral neurons do not make synapses between each other…” Can the authors provide a refence for this statement?

Answers and Actions:

We included references to support this statement. (p:2 l:36-38; p:15 l:375)

Comments to the Authors:

11) Line 363: “Functional and structural synapse formation and network development potential of the cultured sensory neurons may shed new light on the unexplained interactions observed in various disorders of peripheral nervous system.” Which “unexplained interactions” are meant by the authors?

Answers and Actions:

We thank the reviewer for pointing out the ambiguity of this statement. Whole section was revised. (p:15 l:376-383)

Response to Reviewer #2:

Comments to the Authors:

My only major concern is the quality and labeling of the figures. The resolution is too low and labels such as c+ and c- rather than words make the story harder than necessary to follow.

Answers and Actions:

We thank the reviewer for their valuable suggestions. We increased the quality of the figures. We resorted to use definitions c+ and c-, in order to explain the concepts more clearly and to avoid any misinterpretations.

Comments to the Authors:

1. Introduction, line 41: edit "wanted to prove that" to "wanted to examine whether or not"

Answers and Actions:

We implemented this change (p:2 l:43)

Comments to the Authors:

2. Introduction, line 59: edit "transactions" to "transients"

Answers and Actions:

We implemented this change (p:3 l:62)

---

## [Editor Report · Decision Letter 1]

29 Jan 2021

Adult mouse dorsal root ganglia neurons form aberrant glutamatergic connections in dissociated cultures

PONE-D-20-30530R1

Dear Dr. Gülçür,

We’re pleased to inform you that your manuscript has been judged scientifically suitable for publication and will be formally accepted for publication once it meets all outstanding technical requirements.

Kind regards,

Robert Blum

Academic Editor

PLOS ONE
---

## [Editor Report · Acceptance letter]

15 Feb 2021

PONE-D-20-30530R1 

Adult mouse dorsal root ganglia neurons form aberrant glutamatergic connections in dissociated cultures 

Dear Dr. Gülçür:

I'm pleased to inform you that your manuscript has been deemed suitable for publication in PLOS ONE. Congratulations! Your manuscript is now with our production department. 

Kind regards, 

on behalf of

PD Dr. Robert Blum 

Academic Editor

PLOS ONE